# Improving Robustness and Diversity with Adversarial Contrastive Network Ensembles

## Abstract

Relying on ensemble diversity strategies to improve adversarial robustness has been investigated in several papers, but the gains provided by ensemble-based defenses remain limited so far. In this work, we propose Adversarial Contrastive Network (ACN) ensembles as a defense against white-box adversarial attacks which is based on a new ensemble diversity strategy. It consists in projecting the output feature maps of the different ensemble models in a shared latent space with a projection network and using contrastive learning to diversify the feature representations learned by the different models. The performance of the proposed method is evaluated and compared to regular ensembles in terms of adversarial robustness and ensemble diversity. Results obtained demonstrate superior adversarial robustness for ACN ensembles against the Fast Gradient Sign Method attack and against Projected Gradient Descent attacks using low distortion bounds. Lower transferability of adversarial examples among individual models within ACN ensembles is also demonstrated, suggesting that the proposed method helps achieve more diverse representations.

## 1 Introduction

The vulnerability of neural networks against adversarial attacks is well known and has been widely studied over the years, especially in the context of image classification (Szegedy et al., 2014; Goodfellow et al., 2015). In response, many defense mechanisms (e.g. adversarial defenses) have been developed to mitigate the effect of adversarial attacks on the performance of neural networks. Several adversarial defenses proposed in the literature are based on ensembles of neural networks (Pang et al., 2019; Verma & Swami, 2019; Sen et al., 2020; Abbasi & Gagné, 2017; Yang et al., 2020; 2021). These defenses generally aim at improving robustness to adversarial attacks by training ensembles of diverse models using diversity promotion strategies. They are based on the assumption that diverse ensembles are more difficult to fool compared to single networks given that they require adversarial examples to fool multiple models exposing distinct decision behaviors. However, current solutions have a limited level of performance, especially against white-box adversarial attacks. This motivates the search for new strategies to promote ensemble diversity and adversarial robustness against such attacks (Madry et al., 2018; Zhang et al., 2019).

In this work, we propose Adversarial Contrastive Network (ACN) ensembles, an ensemble-based adversarial defense relying on a novel diversity-promoting strategy. Our approach is based on the findings of Ilyas et al. (2019) concerning the fact that different neural networks trained on a similar classification task are likely to learn similar feature representations. Our goal with ACN ensembles is thus to diversify the feature representations learned by the different models forming an ensemble directly from their output feature maps in order to improve the ensemble's adversarial robustness against white-box attacks. For that purpose, ACN ensembles are composed of regular ensembles to which a projection network connected to the output feature maps of the models is added in parallel. While the ensemble is trained to maximize the classification performance of the individual models, ensemble diversity is promoted through the projection network using contrastive learning. Indeed, the projection network is trained with a custom contrastive loss function in order to project the feature maps originating from the models in a shared latent space, where the representations of similar feature maps (i.e., feature maps from images of the same true class label) originating from different models are grouped. This is done to align the representations learned by the different models that are likely

to be similar (Ilyas et al., 2019). A diversity loss based on the distance between the representations of similar feature maps in the shared latent space is then computed and back-propagated in the ensemble models during training to promote the learning of more diverse representations, which cannot be aligned by the projection network. This way, aligning feature maps from the different models should be progressively more difficult to achieve by the projection network. We expect this approach to force individual models to rely on different features to classify images of the same class.

We experimentally evaluate ACN ensembles in terms of adversarial robustness and ensemble diversity and compare their performances with regular ensembles of neural networks. Adversarial robustness is measured with popular white-box adversarial attacks such as Fast Gradient Sign Method (FGSM) and Projected Gradient Descent (PGD). Ensemble diversity is measured with a metric based on the transferability of adversarial examples among individual models in the ensembles. This choice is motivated by the fact that adversarial examples (i.e., images perturbed with adversarial attacks) tend to transfer better between models using similar feature representations (Ilyas et al., 2019). Results obtained expose lower transferability among individual models, suggesting better ensemble diversity and show better robustness against the FGSM attack and PGD attacks of low distortion bounds for ACN ensembles compared to regular ensembles. This shows the potential of ACN ensembles for improving adversarial robustness and as an ensemble diversity strategy.

## 2 Related Work

Some works on ensemble-based adversarial defenses focusing on ensemble diversity have been proposed, the most relevant to the current work being *Adaptive Diversity Promoting* (ADP), *Error Correcting Output Codes* (ECOC), *Ensembles of Mixed Precision Deep Networks for Increased Robustness* (EMPIR), *Diversifying Vulnerabilities for Enhanced Robust Generation of Ensembles* (DVERGE) and *Transferability Reduced Ensemble via Promoting Gradient Diversity and Model Smoothness* (TRS).

**ADP** is an adversarial defense relying on a specific loss to promote diversity (Pang et al., 2019). With this method, the $M$ ensemble models are trained with the following loss function:

$$L(\boldsymbol{x}, y) = -\alpha \, \mathcal{H}(\boldsymbol{z}(\boldsymbol{x})) - \beta \, \mathrm{Vol}^2 \left( \{ \boldsymbol{z}_m^{\backslash y}(\boldsymbol{x}) \} \right) + \sum_{m=1}^{M} L_{CE}(\boldsymbol{z}_m(\boldsymbol{x}), y),$$

where $\boldsymbol{z}_m(\boldsymbol{x})$ is the $m$-th model probability vector output, $\boldsymbol{z}(\boldsymbol{x})$ is the ensemble probability output, $L_{CE}(\boldsymbol{z}_m(\boldsymbol{x}), y)$ is the cross-entropy loss of the $m$-th model, $H(\boldsymbol{z}(\boldsymbol{x}))$ is the Shannon entropy of the ensemble, and $\alpha$ and $\beta$ are hyperparameters. The middle term (i.e., $\mathrm{Vol}^2(\cdot)$) is the volume spanned by the probability vectors of the non-maximal predictions $\{ \boldsymbol{z}_m^{\backslash y}(\boldsymbol{x}) \}$, which includes the probabilities for all classes except that of class $y$. This term is introduced since its value can only increase if the vectors of non-maximal predictions from the different models are diversified. Consequently, this strategy aims at diversifying the output probability vectors of the different models while having a minimal impact on the ensemble's accuracy since the maximal predictions are excluded from the volume-based regularization term.

**ECOC** ensembles were initially proposed by Dietterich & Bakiri (1995) and later adopted as an adversarial defense by Verma & Swami (2019). They are based on error correction methods used in digital communication to control errors in data transmission over unreliable or noisy communication lines. For a classification

Table 1: Example of a codeword matrix for classifying over $C = 4$ classes using $M = 10$-bit class codewords.

| Class | Binary classifiers | | | | | | | | | |
|---|---|---|---|---|---|---|---|---|---|---|
| | $f_1$ | $f_2$ | $f_3$ | $f_4$ | $f_5$ | $f_6$ | $f_7$ | $f_8$ | $f_9$ | $f_{10}$ |
| $C_1$ | -1 | -1 | -1 | +1 | +1 | -1 | -1 | -1 | -1 | +1 |
| $C_2$ | +1 | -1 | -1 | -1 | -1 | -1 | +1 | -1 | +1 | -1 |
| $C_3$ | +1 | -1 | +1 | +1 | +1 | -1 | +1 | +1 | +1 | -1 |
| $C_4$ | -1 | -1 | +1 | -1 | +1 | +1 | -1 | +1 | -1 | -1 |

problem of $C$ classes, ECOC ensembles are composed of $M$ binary classifiers trained over different subsets of classes represented by the columns of a $C \times M$ matrix, such as the example shown in Table 1. In such a matrix, the rows represent the $M$-bit class codewords assigned to each class. At inference, the output bits of the binary classifiers are concatenated to form codewords compared to the class codewords using the Hamming distance to determine the inferred classes. ECOC ensembles have interesting robustness properties when the matrix is carefully designed. For instance, if a minimal Hamming distance of $\theta$ is enforced between any pairs of class codewords, the ensemble will be robust to wrong predictions of $\lfloor \frac{\theta-1}{2} \rfloor$ binary classifiers. Moreover, ensemble diversity can be promoted by maximizing the column separation in the codeword matrix, to diversify the subsets of classes the binary classifiers are trained on.

**EMPIR** is an adversarial defense proposed by Sen et al. (2020) and based on ensembles of Quantized Neural Networks (i.e., networks storing weights and activations in variables of varying precisions). EMPIR ensembles are composed of a network with 32-bit weights and activations, a network with 2-bit weights and 4-bit activations, and a network with 2-bit weights and activations. Final predictions are made according to majority voting over the class predictions of each model. According to the authors, EMPIR ensembles are less sensitive to adversarial perturbations.

**DVERGE** is an adversarial defense proposed by Yang et al. (2020) which is based on diversifying the adversarial vulnerabilities of the different neural networks in an ensemble. The authors proposed a new loss function that maximizes the diversity of every pair of models based on their non-robust features using their feature maps. Although DVERGE also uses the feature maps to diversify the ensembles, it is much different from our approach, which aims at diversifying the feature representations through contrastive learning with a projection network.

**TRS** is an adversarial defense proposed by Yang et al. (2021) based on an extensive theoretical evaluation of ensemble diversity and robustness. The authors found that enforcing model smoothness as well as gradient orthogonality can help reduce the transferability of adversarial examples among the models within an ensemble, and thus the adversarial robustness of the ensemble. They designed a new training objective function based on these two conditions to train more diverse ensembles.

The ensemble-based defenses presented use different strategies to promote ensemble diversity and adversarial robustness. More recent work such as DVERGE and TRS achieve good results in terms of reducing adversarial transferability of adversarial examples among ensemble models. However, their robustness against white-box adversarial attacks is still limited, which motivates the research for new ensemble diversity strategies to improve adversarial robustness to such attacks.

To the best of our knowledge, this marks one of the first instances where contrastive learning is employed with ensembles of neural networks within the context of adversarial robustness. Previous studies on the intersection of contrastive learning and adversarial robustness, with frameworks such as *Robust Contrastive Learning* (RoCL) (Fan et al., 2021) and *Adversarial Contrastive Learning* (AdvCL) (Kim et al., 2020), have focused on improving the robustness of self-supervised learning models trained with unlabeled data.

## 3 Methodology

In the following, we introduce ACN ensembles as a novel ensemble diversity strategy to improve adversarial robustness. This differs from previous work on ensemble diversity by acting directly on the feature representations of the models using a projection network and contrastive learning.

### 3.1 Architecture of ACN Ensembles

Fig. 1 presents the overall architecture of ACN ensembles in the context of image classification. As shown, input images are classified with $M$ models (i.e., neural networks). Each model in the ensemble is composed of a set of feature extraction layers (i.e., $f_m$) and a set of classification layers (i.e., $g_m$). As shown in Fig. 1 the feature map $\boldsymbol{q}_m = f_m(\boldsymbol{x})$ is extracted by the $m$-th model from the input image $\boldsymbol{x}$, with $\boldsymbol{z}_m = g_m(\boldsymbol{q}_m)$ being the corresponding output classification probability vector. The voting mechanism used to generate the ensemble probability vector (i.e., $\boldsymbol{z}_{\text{tot}}$) is shown in the "Voting and prediction" block of Fig. 1. It consists

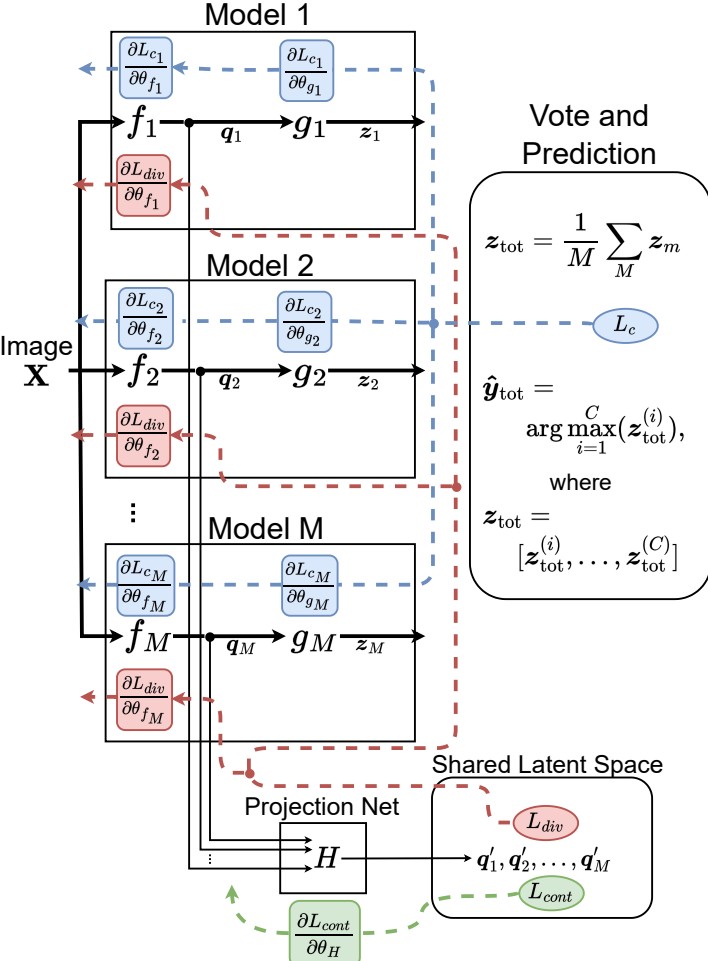

Figure 1: Architecture of ACN ensembles. The ensemble is composed of $M$ neural networks and a voting mechanism for image classification. The neural networks are decomposed into two components: $f$ representing the feature extraction layers and g representing the classification layers. A projection network $H$, situated in parallel to the ensemble, consists of fully connected layers followed by a ReLU activation function. It is responsible for projecting the feature maps $q$ at the output of the feature extractors into the representations $q'$ in the shared latent space. The neural networks are trained to minimize the classification loss $L_c$ represented in blue, and the projector is trained to minimize the contrastive loss $L_{cont}$ represented in green. The diversity loss $L_{div}$ in red is calculated on the representations in the shared latent space and is backpropagated in the feature extraction layers to diversify the features learned by different models.

in a linear combination of the probability vectors (i.e., $z_m$) of all models. This voting mechanism allows the ensemble to be fully differentiable, allowing the generation of white-box adversarial examples (Tramèr et al. 2020). Finally, the class prediction $\hat{y}$ of an image $x$ is determined according to the highest class probability among the $C$ classes of $z_{tot}$. The models are trained with loss function $L_c$, shown in blue in Fig. 1 and discussed further in Sec. 3.3.

The projection network $H$ is operating in parallel to the ensemble. As shown in Fig. 1, a set of fully connected layers process the output feature maps of the $M$ feature extractors to project them in a shared latent space, that is $q'_m = H(q_m)$. Note that the feature maps are normalized to the unit hypersphere before being processed by the projection network (Chen et al., 2020; Khosla et al., 2020). The goal of the projection network is to represent the feature maps originating from the individual models in a new latent space where

representations of similar feature maps are grouped and those of different feature maps separated. We define similar feature maps as those coming from different models and images of the same class. As for the different feature maps, they are defined as those coming from different models and images of different classes. The projection network is trained with a custom contrastive loss function $L_{cont}$, shown in green in Fig. 1 and presented in Sec. 3.3.

## 3.2 Diversity Promotion

Explicit diversity promotion is required in order for the $M$ models of an ensemble to learn different feature representations. Otherwise, different neural networks trained on the same classification task tend to learn similar feature representations leading to the models having similar adversarial vulnerabilities (Ilyas et al. 2019; Pang et al. 2019). With ACN ensembles, the feature extraction layers of the $M$ models are regularized using the representations in the shared latent space defined by the projection network. We use these representations to evaluate the similarity of the feature representations learned and to foster the learning of different features to classify images of the same class. This is inspired by the domain classifier network used in Domain Adversarial Neural Networks (DANN) to promote the learning of domain invariant features in neural networks (Ganin et al. 2016).

The inclusion of the projection network in ACN ensembles is justified by the fact that it enables us to assess the similarity of the feature representations learned by the different models. We assume that if the projection network is able to project the feature maps of the different models in a shared space where they are grouped according to their true class labels, this is likely due to the fact that the models are relying on similar feature representations. Therefore, to achieve more diverse representations, the feature extractors of the $M$ models are regularized during training such that it will become progressively more difficult for the projection network to correctly align their feature maps in the shared latent space. This is achieved through a competition between the contrastive and diversity losses. The contrastive loss $L_{cont}$ is minimized when training the projection network to align the representations of similar feature maps. Conversely, the diversity loss $L_{div}$, shown in red in Fig. 1, aims at separating similar feature maps in the shared latent space, thus acting in opposition to the alignment task of the contrastive loss with the projection network. While the contrastive loss is used to directly train the projection network, the diversity loss is backpropagated into the feature extractors of the individual models, acting as a regularization factor for the classification loss.

---

**Algorithm 1** Training algorithm of ACN ensembles

**Inputs:** Batch size $N$, number of models $M$, epochs $E$, feature extractors $f_{1,\dots,M}$, classifiers $g_{1,\dots,M}$, projection network $H$, hyperparameters $\lambda$, $\alpha_1$ and $\alpha_2$, training data $\mathcal{X}$ and corresponding labels $\mathcal{Y}$

**for** epoch $e = 1, \dots, E$ **do**

    **for all** mini-batches $\bar{\mathcal{X}} \subset \mathcal{X}$ **do**

        $\boldsymbol{q}_m^{(i)} = f_m(\boldsymbol{x}_i),\ m = 1, \dots, M,\ \forall \boldsymbol{x}^{(i)} \in \bar{\mathcal{X}}$         ▷ Extract the output feature maps

        $\boldsymbol{z}_m^{(i)} = g_m(\boldsymbol{q}_m^{(i)}),\ m = 1, \dots, M,\ \forall \boldsymbol{x}^{(i)} \in \bar{\mathcal{X}}$         ▷ Classification probability vectors

        $\boldsymbol{q}_m'^{(i)} = H(\boldsymbol{q}_m^{(i)}),\ m = 1, \dots, M,\ \forall \boldsymbol{x}^{(i)} \in \bar{\mathcal{X}}$         ▷ Representations in the shared latent space

        ▷ Compute $L_c$, $L_{cont}$, $L_{div}$ (Eq. 1, 2 and 3)

        ▷ Update parameters with gradient descent

        $\theta_{g_m} \leftarrow \theta_{g_m} - \alpha_1 \frac{\partial L_c}{\partial \theta_{g_m}},\ m = 1, \dots, M$         ▷ Classification layers with $L_c$

        $\theta_{f_m} \leftarrow \theta_{f_m} - \alpha_1 \left( \frac{\partial L_c}{\partial \theta_{f_m}} + \lambda \frac{\partial L_{div}}{\partial \theta_{f_m}} \right),\ m = 1, \dots, M$     ▷ Feature extractors with $L_c$ and $L_{div}$

        $\theta_H \leftarrow \theta_H - \alpha_2 \frac{\partial L_{cont}}{\partial \theta_H}$         ▷ Projection network with $L_{cont}$

    **end for**

**end for**

---

### 3.3 Training of ACN Ensembles

The ACN training procedure is shown in Algorithm 1. It begins by processing a mini-batch set $\bar{\mathcal{X}}$ and corresponding labels $\bar{\mathcal{Y}}$ sampled from the complete training set $\mathcal{X}$ with labels $\mathcal{Y}$. The images of the mini-batch are first passed through the $M$ models and the projection network to obtain the feature maps, their classification probability vectors and their representations in the shared latent space. Then, the three losses $L_c$, $L_{cont}$, and $L_{div}$ are computed over the mini-batch samples to update the weights of the feature extractors, classifiers and projection network. This process is repeated for all mini-batches over the $E$ epochs.

Loss function $L_c$ is used to simultaneously train the $M$ models of the ensemble on the classification task:

$$L_c(\theta_f, \theta_g, \bar{\mathcal{X}}, \bar{\mathcal{Y}}) = \frac{1}{|\bar{\mathcal{X}}|M} \sum_{\boldsymbol{x}^{(i)} \in \bar{\mathcal{X}}} \sum_{m=1}^{M} L_{\text{CE}}(\theta_{f_m}, \theta_{g_m}, \boldsymbol{z}_m^{(i)}, y^{(i)}), \tag{1}$$

where $L_{\text{CE}}$ is the cross-entropy loss of the $m$-th model. As the individual cross-entropy losses are summed, this training method is similar to training each model independently since each model has its own independent set of parameters (Pang et al., 2019). Simultaneous training is important for the proposed method because it allows the ensemble to be regularized based on the interactions between the models during training (Islam et al., 2003).

The projection network $H$ is trained with contrastive learning. This type of learning is typically used to learn data representations in a new latent space based on the contrast between representations of similar and dissimilar data (Chen et al., 2020). A contrastive network is usually composed of fully connected layers connected to the feature extraction layers of a neural network and it generates vector representations of the images in the new latent space (Chen et al., 2020; Khosla et al., 2020). In addition, both feature extraction layers and the contrastive network are usually trained with a contrastive loss function to maximize the alignment of the representations of similar data in the new latent space. Inspired by previous contrastive learning works (Sen et al., 2020; Khosla et al., 2020), the projection network $H$ in ACN ensembles generates vector representations of the feature maps extracted from the different models in what we define as the shared latent space. It is trained with the objective of grouping together representations of similar images (same ground truth labels) and separating representations of different images (different ground truth labels) using the following contrastive loss:

$$L_{cont}(\theta_H, \bar{\mathcal{X}}, \bar{\mathcal{Y}}) = \frac{1}{|\bar{\mathcal{X}}|M} \sum_{\boldsymbol{x}^{(i)} \in \bar{\mathcal{X}}} \sum_{m=1}^{M} \frac{-1}{|\mathcal{Q}_{\setminus m}^{\prime(i)}|} \sum_{\boldsymbol{p} \in \mathcal{Q}_{\setminus m}^{\prime(i)}} \log \frac{\exp(\text{sim}(\boldsymbol{q}_m^{\prime(i)}, \boldsymbol{p})/\tau)}{\sum_{\boldsymbol{a} \in \{\mathcal{A}_{\setminus m}^{\prime(i)} \cup \boldsymbol{p}\}} \exp(\text{sim}(\boldsymbol{q}_m^{\prime(i)}, \boldsymbol{a})/\tau)}, \tag{2}$$

where $\text{sim}(\cdot)$ is the cosine similarity, the metric used to measure the distance between representation pairs in the shared latent space, and $\tau$ is the temperature parameter used for calibration (Guo et al., 2017). Let us define $\mathcal{Q}_m' = \{\boldsymbol{q}_m^{\prime(i)} \,|\, \boldsymbol{x}^{(i)} \in \bar{\mathcal{X}}\}$ as the set of representations in the shared latent space of the feature maps corresponding to the instances $\boldsymbol{x}^{(i)} \in \bar{\mathcal{X}}$ and originating from the $m$-th model. We then define set $\mathcal{Q}_{\setminus m}^{\prime(i)} = \{\boldsymbol{q}_l^{\prime(j)} \in \mathcal{Q}_l', \forall l \neq m \,|\, y^{(j)} = y^{(i)}\}$, which consists of the representations in the shared latent space of the feature maps corresponding to the mini-batch instances $\boldsymbol{x}^{(j)}$ having the same ground truth label as $\boldsymbol{x}^{(i)}$ and originating from all models other than $m$. We also define set $\mathcal{A}_{\setminus m}^{\prime(i)} = \{\boldsymbol{q}_l^{\prime(j)} \in \mathcal{Q}_l', \forall l \neq m \,|\, y^{(j)} \neq y^{(i)}\}$ as the representations in the shared latent space of the feature maps corresponding to the mini-batch instances originating from all models other than $m$ and having ground truth labels which differ from that of $\boldsymbol{x}^{(i)}$. With this formulation of $L_{cont}$, the cosine similarity between pairs of representations of similar feature maps is maximized and the cosine similarity between pairs of representations of different feature maps is minimized in the shared latent space. Note that the sum at the denominator of Eq. 2 follows a similar form as the SimCLR loss introduced by Chen et al. (2020). The effect of the contrastive loss is shown in Fig. 2(a).

The final loss function used for ACN ensembles training is the diversity loss $L_{div}$. This loss function defined as

$$L_{div}(\theta_f, \bar{\mathcal{X}}, \bar{\mathcal{Y}}) = \frac{1}{|\bar{\mathcal{X}}|M} \sum_{\boldsymbol{x}^{(i)} \in \bar{\mathcal{X}}} \sum_{m=1}^{M} \frac{1}{|\mathcal{Q}_{\setminus m}^{\prime(i)}|} \sum_{\boldsymbol{p} \in \mathcal{Q}_{\setminus m}^{\prime(i)}} \text{sim}(\boldsymbol{q}_m^{\prime(i)}, \boldsymbol{p}), \tag{3}$$

is used to regularize the feature extraction layers of the models and diversify their feature representations. In the equation, set $\mathcal{Q}_{\backslash m}^{\prime(i)}$ is the same set defined earlier for Eq. 2, containing the representations in the shared latent space of the feature maps corresponding to the mini-batch instances $\boldsymbol{x}^{(j)}$ having the same ground truth label as $\boldsymbol{x}^{(i)}$ and originating from all models other than $m$. Loss $L_{div}$ corresponds to the average cosine similarity between all pairs of similar representations in the shared latent space. The parameters of the feature extractors are updated with backpropagation in a way that minimizes $L_{div}$, to promote the separation of the similar representations in the shared latent space.

Since these representations are from feature maps originating from different models and from images with the same true class labels, this regularization of the feature extractors will promote the diversification of their representations by forcing them to use more diverse features to classify images of the same class. Fig. 2(b) helps visualize the effect of the diversification loss on the representations in the shared latent space. It will also make the training of the projection network progressively more difficult since $L_{div}$ is antagonistic to $L_{cont}$. Ultimately, the goal is attained if the regularization with the diversity loss leads to an increase of $L_{cont}$, meaning that the feature representations are diversified and the projection network can no longer group the similar feature maps originating from different models in the shard latent space.

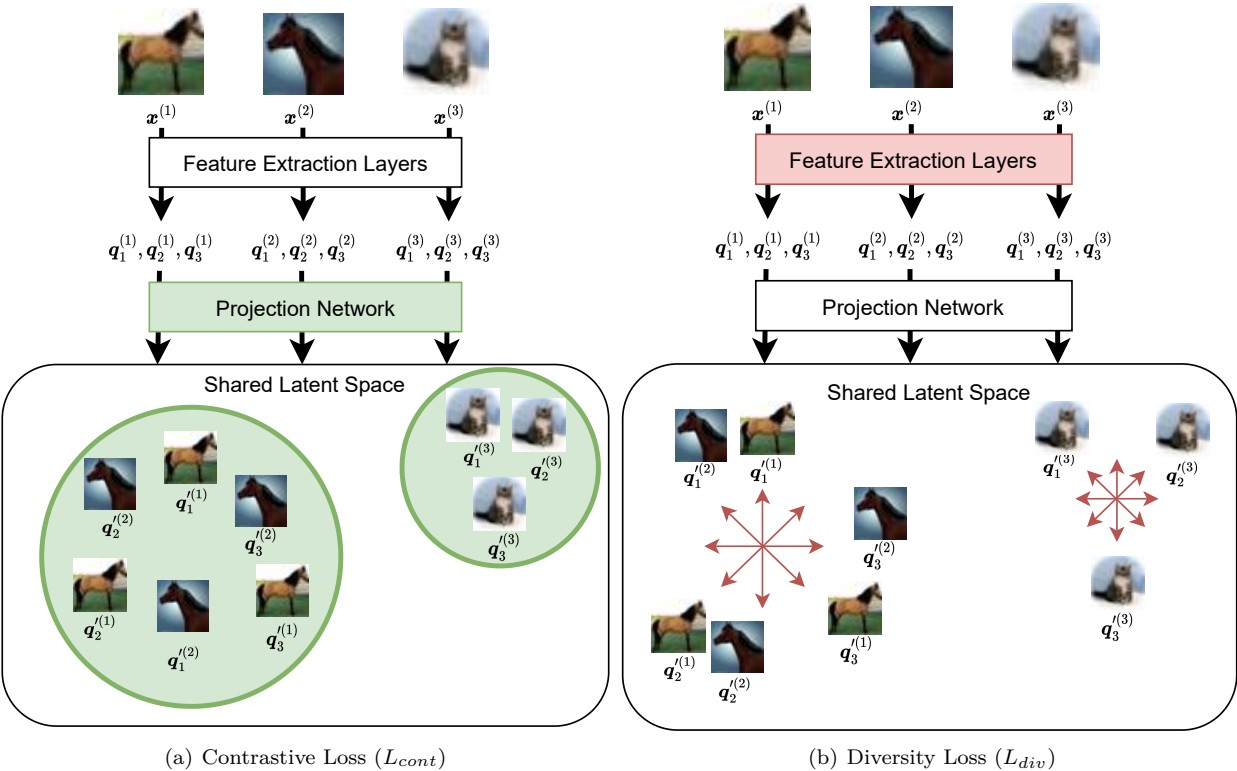

(a) Contrastive Loss ($L_{cont}$)    (b) Diversity Loss ($L_{div}$)

Figure 2: High level visualization of the impact of $L_{cont}$ and $L_{div}$ on the representations $\boldsymbol{q}'$ in the shared latent space. In this example, we assume an ACN ensemble composed of $M = 3$ models. Therefore, each input image $\boldsymbol{x}^{(i)}$ has 3 feature maps $\boldsymbol{q}_m^{(i)}$ and 3 corresponding representations $\boldsymbol{q}_m'^{(i)}$ generated at the output of the projection network. As shown in (a), the contrastive loss impacts only the projection network during the training and its goal is to gather the representations $\boldsymbol{q}'$ coming from different models and from images of the same class (e.g. $\boldsymbol{q}_1'^{(3)}$ and $\boldsymbol{q}_3'^{(3)}$). The effect of the diversity loss is shown in (b). It impacts the feature extraction layers and its goal is to separate the representations of images of the same class and coming from different models (e.g. $\boldsymbol{q}_2'^{(1)}$ and $\boldsymbol{q}_1'^{(1)}$). The representations of images of the same class and coming from the same model like $\boldsymbol{q}_2'^{(1)}$ and $\boldsymbol{q}_1'^{(1)}$ are not separated in (b) because we assume the model should extract similar features for such images.

The gradient descent update equations for all components ($\theta_{g_m}$, $\theta_{f_m}$ and $\theta_H$) are shown at the bottom of Algorithm 1. The weights of the classification layers are updated to minimize the loss $L_c$ and the weights of the projection network are updated to minimize the loss $L_{cont}$. The weights of the feature extraction layers are updated to minimize both $L_c$ and $L_{div}$ in order to extract meaningful and diverse features to achieve good accuracy and ensemble diversity. Parameters $\alpha_1$ and $\alpha_2$ in the update equations correspond to the ensemble and projection network learning rates, respectively. Finally, $\lambda$ is the diversification parameter, allowing the weight of $L_{div}$ to be controlled in the update of the parameters of the feature extraction layers. This parameter will be optimized in the experiment since a value which is too high could prevent the projection network from learning proper mappings in the shared latent space.

## 4    Experiments

We conducted several experiments to demonstrate the proper functioning of the proposed method and to show the improvements in robustness and ensemble diversity for ACN ensembles compared to regular ensembles of neural networks. The experiments are performed with the Fashion-MNIST (Xiao et al., 2017) and CIFAR10 (Krizhevsky & Hinton, 2009) datasets. ACN ensembles trained on Fashion-MNIST consist of three ($M = 3$) ResNet20 (He et al., 2016) and a projection network composed of three fully connected layers of size 64, 64 and 32, each followed by ReLU activations. Each feature extractor is thus composed of the input convolutional layer and the residual blocks over which the classification is carried out by a fully connected layer of size 64. The ACN ensembles trained on CIFAR10 consist of three ($M = 3$) ResNet18 (He et al., 2016) and a projection network composed of three fully connected layers of size 512, 256 and 64, each followed by ReLU activations. The feature extractors and classifiers have similar architectures to the ACN ensembles trained on Fashion-MNIST, except that the feature maps extracted by the ResNet18 are of size 512 instead of 64. The baseline ensembles consist of three ResNet20 for Fashion-MNIST and three Resnet18 for CIFAR10. They are trained simultaneously with the loss function in Eq. 1. Preliminary experiments were conducted to compare the performance of different architectures and motivate the choices made (see Appendix A for more details).

In the experiments, unless stated otherwise, the ensembles are trained over 1000 epochs with a mini-batch size of $N = 100$, learning rates of $\alpha_1$ and $\alpha_2$ set to 0.001 and a temperature parameter $\tau$ of 0.07. These parameters were selected to optimize the classification and robustness performances. Also, 1000 images from the training data are randomly selected and set aside during training to evaluate the performance in validation. Furthermore, 2000 images randomly sampled from the test dataset are used to measure the performance in test after training.

### 4.1    Projection Network

The goal of this experiment is to show that the contrastive loss function proposed in Eq. 2 is effective for training the projection network. That is, it allows the projection network to learn to project the feature maps coming from the different models into the shared latent space, as described in Sec. 3.1. For that purpose, the dimensionality reduction method t-SNE (van der Maaten & Hinton, 2008) is used to visualize the feature maps (i.e., $\boldsymbol{q}$) at the output of the feature extractors and their representations in the shared latent space (i.e., $\boldsymbol{q}'$) at the output of the projection network. For this experiment, the diversity loss $L_{div}$ is disabled during training of the ACN ensemble to allow the projection network and the ensemble to be trained independently, without diversity promotion.

The results are shown in Fig. 3, where we can see on the left the feature maps at the output of the feature extractors and on the right their representations in the shared latent space generated by the projection network. We observe in Fig. 3(a) that the feature maps are grouped by true class labels (marker color) as well as by feature extractor of origin (marker type). This results in 3 main groups for the 3 feature extractors in the ACN ensemble, each composed of 10 clusters for the 10 classes. In contrast, we observe in Fig. 3(b) that the representations originating from different feature extractors (different marker types) and from images of the same class (same color) are grouped in clusters. Although the clusters for the 10 classes are not perfectly grouped in Fig. 3(b), we can see that the projection network managed to align the feature representations originating from different feature extractors and images of the same true class

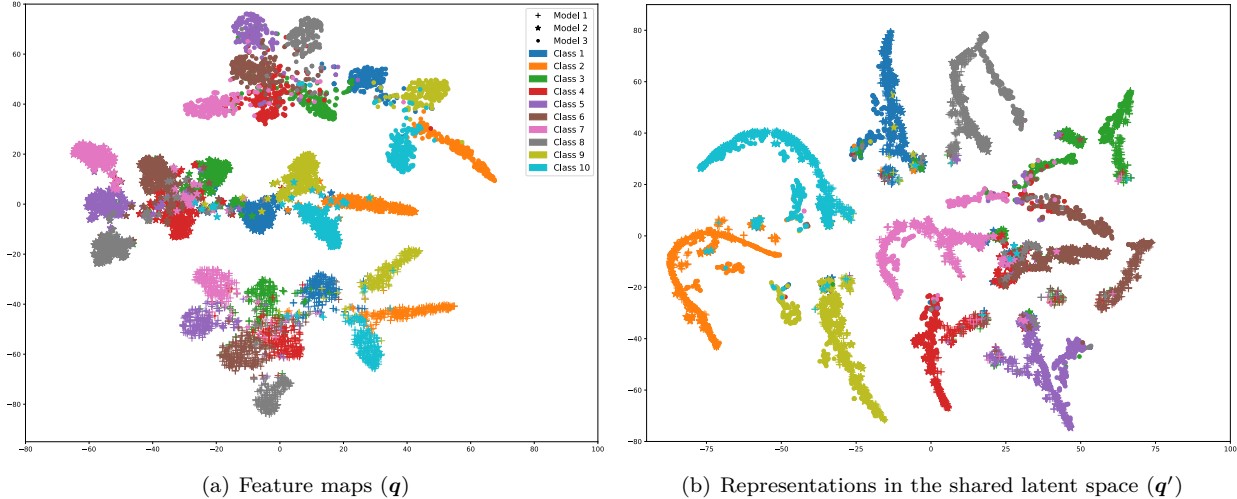

(a) Feature maps ($\boldsymbol{q}$)

(b) Representations in the shared latent space ($\boldsymbol{q'}$)

Figure 3: t-SNE visualization of 2000 feature maps $\boldsymbol{q}$ at the output of the feature extractors and their corresponding representations $\boldsymbol{q'}$ in the shared latent space generated by the projection network. The marker color represents the true class label and the marker type the model from which the feature maps and the representations are originating. The ACN ensemble is trained on CIFAR10 for 50 epochs since the contrastive loss $L_{cont}$ becomes stable after this point.

label pretty well compared to their original representations in Fig. 3(a). This shows that the proposed contrastive loss works as intended for training the projection network. This experiment was also performed with Fashion-MNIST (see Appendix B).

### 4.1.1 Robustness

We now aim at demonstrating experimentally that ACN ensembles can support the learning of more diverse features, which in turn improves robustness to adversarial attacks in comparison to baseline ensembles. In this section, we assess the robustness of ACN ensembles compared to the baseline ensembles. Non-targeted white-box adversarial attacks are used to evaluate robustness, which is measured as the accuracy of the ensembles against the generated adversarial examples. The attacks used are FGSM and PGD from the CleverHans V4.0.0 library (Papernot et al., 2018) and the adversarial perturbations are bounded by an $\ell_\infty$ norm of 0.1 for Fashion-MNIST and 0.031 for CIFAR10. PGD attacks are performed with 20 iterations and a gradient step of $\frac{\ell_\infty}{3}$.

As mentioned in sec. 3.3, the parameter $\lambda$ controls the weight of the diversification loss when updating the parameters of the feature extraction layers. Several ACN ensembles were trained on both datasets with different values of $\lambda$ to optimize this parameter and study its effect on robustness. The values 0.01, 0.02, 0.04 and 0.06 were tested with Fashion-MNIST and the values 0.025, 0.05, 0.075 and 0.1 were tested with CIFAR10. ACN ensembles are henceforth labeled F-ACN$_\lambda$ and C-ACN$_\lambda$ according to the parameter $\lambda$ and the dataset used (F for Fashion-MNIST and C for CIFAR10). The baseline ensembles are labeled F-Vanilla and C-Vanilla.

Fig. 4 shows the robustness against the FGSM attack computed from the validation images at different epochs during training for the ACN and baseline ensembles. This figure shows that ACN ensembles can be more robust to the FGSM attack than regular ensembles. It also shows the importance of tuning $\lambda$. Indeed, we notice that increasing the value of $\lambda$ eventually leads to a drop in robustness. For instance, a drop is observed in Fig. 4(a) between F-ACN$_{0.02}$ and F-ACN$_{0.04}$ and between C-ACN$_{0.075}$ and C-ACN$_{0.1}$ in Fig. 4(b). This is probably due to the diversity loss having too much importance in the training of the feature extractors, which causes a rapid increase in the contrastive loss indicating that the projection network is rapidly fooled and unable to properly map the feature maps and adapt to the changes in the features learned

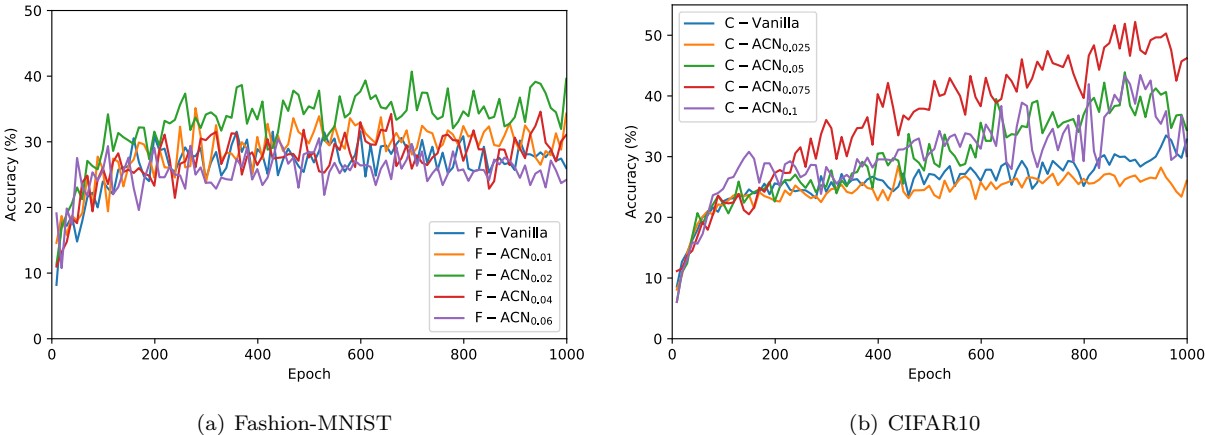

(a) Fashion-MNIST                                    (b) CIFAR10

Figure 4: Robustness against FGSM at different epochs during training for the baseline and different ACN ensembles trained on Fashion-MNIST and CIFAR10. Robustness is evaluated every 10 epochs with the validation set.

Table 2: Test accuracy and robustness against FGSM and PGD attacks for ACN ensembles trained on Fashion-MNIST and CIFAR10 compared to the baseline ensembles. Adversarial examples are generated from the test images and the perturbations are bounded by a $\ell_\infty$ norm of 0.1 for Fashion-MNIST and 0.031 for CIFAR10.

| Model | Clean Accuracy (%) | | | | Robustness (%) | |
|---|---|---|---|---|---|---|
| | Ensemble | Net 1 | Net 2 | Net 3 | FGSM | PGD |
| F-Vanilla | 95.0 | 94.2 | 94.5 | 94.1 | 31.0 | 0.0 |
| F-ACN$_{0.02}$ | **95.8** | **95.5** | **94.8** | **95.1** | **40.6** | **0.2** |
| C-Vanilla | **95.7** | **94.3** | **94.05** | **93.9** | 32.8 | **8.6** |
| C-ACN$_{0.075}$ | 94.5 | 93.1 | 93.6 | 93.2 | **52.2** | 6.9 |

(See Appendix C for more details). F-ACN$_{0.02}$ and C-ACN$_{0.075}$ will be used in the next experiments since they are the most robust ACN ensembles against the FGSM attack in Fig. 4.

In Table 2, F-ACN$_{0.02}$ and C-ACN$_{0.075}$ are compared to F-Vanilla and C-Vanilla in terms of test accuracy on clean images and robustness against different adversarial attacks. The ensembles used to generate the results were selected at the training epoch where their validation FGSM robustness is the highest to promote their individual robustness. Looking at the results in Table 2, we notice that ACN ensembles have similar accuracy scores as the baselines, which shows that the proposed diversification strategy does not affect classification performances. In terms of robustness against FGSM, the results show increases of about 10% for F-ACN$_{0.02}$ and about 20% for C-ACN$_{0.075}$ compared to the baselines. This shows the superiority of ACN ensembles against the FGSM attack. Nevertheless, the results in Table 2 are not as conclusive for PGD, as the robustness of all ensembles against this attack remain low.

To further investigate robustness against the PGD attack, Tables 3 and 4 show the robustness of the ensembles against this attack when the adversarial perturbations are bounded by different $\ell_\infty$ norms. We observe in Table 3 that C-ACN$_{0.075}$ has a robustness of 39.65% against PGD bounded by a $\ell_\infty$ norm of 0.01 while the robustness of C-Vanilla against this attack is only 16.9%. Similarly, the robustness of F-ACN$_{0.02}$ against PGD bounded by a $\ell_\infty$ norm of 0.03 in Table 4 is 25.7% while that of F-Vanilla is only 9.7%. These results demonstrate the superiority of ACN ensembles in terms of adversarial robustness against PGD attacks bounded by small $\ell_\infty$ norms.

Table 3: Test Robustness of an ACN ensemble and a baseline ensemble trained on CIFAR10 against PGD attacks bounded by different $\ell_\infty$ norms.

| Model | PGD Robustness (%) | | |
|---|---|---|---|
| | $\ell_\infty = 0.031$ | $\ell_\infty = 0.02$ | $\ell_\infty = 0.01$ |
| C-Vanilla | **8.6** | 11.8 | 16.9 |
| C-ACN$_{0.075}$ | 6.9 | **15.7** | **39.65** |

Table 4: Test Robustness of an ACN ensemble and a baseline ensemble trained on Fashion-MNIST against PGD attacks bounded by different $\ell_\infty$ norms.

| Model | PGD Robustness (%) | | |
|---|---|---|---|
| | $\ell_\infty = 0.1$ | $\ell_\infty = 0.06$ | $\ell_\infty = 0.03$ |
| F-Vanilla | 0.0 | 0.2 | 9.7 |
| F-ACN$_{0.02}$ | **0.2** | **2.5** | **25.7** |

### 4.1.2 Diversity

In this section, we assess ensemble diversity in ACN ensembles. The notion of ensemble diversity is generally based on the assumption that different neural networks commit errors on different data points (Kuncheva, 2004). For this reason, many ensemble architectures are composed of weaker neural networks, to promote diversity in their output predictions (Kuncheva, 2004; Alpaydin, 2010). In our case, ACN ensembles are composed of high-performance models (cf. individual accuracy scores in Table 2). Consequently, traditional ensemble diversity metrics based on output probabilities are not appropriate for measuring ensemble diversity.

To measure diversity, we decided to use a metric based on the transferability of adversarial examples among individual ensemble models. Our working assumption is that the transferability of adversarial examples should be lower between neural networks using more diverse representations since it has been shown that adversarial examples transfer better among models using similar features representations (Ilyas et al., 2019). Therefore, to measure diversity in an ensemble of $M$ neural networks, adversarial examples are generated on each model and the robustness scores of the $M-1$ other models are evaluated against them. This results in a total of $M \times (M-1)$ robustness scores, named transfer robustness scores. Following our assumption, the more diverse the neural networks are, the higher the transfer robustness scores should be because it would mean that less adversarial examples transfer between them. We defined as *average transfer robustness* a metric evaluating the ensemble diversity by averaging these $M \times (M-1)$ transfer robustness values.

The results of the average transfer robustness for F-ACN$_{0.02}$ and C-ACN$_{0.075}$ in comparison to the baseline ensembles are shown in Table 5. The ensembles used for generating these results are the same as in Table 2. We notice that the average transfer robustness for F-ACN$_{0.02}$ is 8% and 7.9% higher compared to F-Vanilla

Table 5: Average transfer robustness against FGSM and PGD attacks for ACN ensembles trained on Fashion-MNIST and CIFAR10 compared to regular ensembles. Adversarial examples are bounded by a $\ell_\infty$ norm of 0.1 for Fashion-MNIST and 0.031 for CIFAR10.

| Model | Average Transfer Robustness (%) | |
|---|---|---|
| | FGSM | PGD |
| F-Vanilla | 32.3 | 15.4 |
| F-ACN$_{0.02}$ | **40.3** | **23.3** |
| C-Vanilla | 41.1 | 16.9 |
| C-ACN$_{0.075}$ | **58.2** | **24.0** |

Table 6: Test accuracy and robustness against the FGSM and PGD attacks for an ACN ensemble and a regular ensemble trained on CIFAR10 with adversarial training. The adversarial examples are generated from the test images and the perturbations are bounded by a $\ell_\infty$ norm of 0.031.

| Model | Clean Accuracy (%) | | | | Robustness (%) | |
|---|---|---|---|---|---|---|
| | Ensemble | Net 1 | Net 2 | Net 3 | FGSM | PGD |
| ADVT-Vanilla | **84.9** | 81.8 | **82.9** | 81.8 | 59.5 | **45.8** |
| ADVT-ACN$_{0.2}$ | 84.3 | **82.2** | 82.5 | **82.7** | **63.8** | 45.4 |

against FGSM and PGD, respectively. For C-ACN$_{0.075}$, these values are respectively 17.1% and 7.1% higher than those of C-Vanilla. These results suggest that the individual models in ACN ensembles learn more diverse representations, since their average transfer robustness is higher, meaning that the transferability of adversarial examples among the models is lower than for the baseline ensembles.

## 5 Adversarial Training

In this last experiment, we evaluate the combination of the proposed method with adversarial training, which is considered one of the most effective adversarial defenses (Croce & Hein, 2020; Rice et al., 2020; Cohen et al., 2019). The adversarial training method used in this experiment is based on the approach described by Madry et al. (2018) that we adapted for ensembles of neural networks. That is, at every epoch during training, adversarial examples are generated for each individual model from the images in the mini-batch. The images generated for each model are then used to compute their individual cross-entropy loss, which are then used to determine the ensemble loss with Eq. 1. The PGD attack is used during training with an $\ell_\infty$ norm of 0.031, 5 steps and a step size of $\frac{\ell_\infty}{3}$.

The results for the ACN ensemble (ADVT-ACN$_{0.2}$) and the baseline ensemble (ADVT-Vanilla) combined with adversarial training are shown in Table 6. We observe that ACN ensembles can be combined with adversarial training without affecting the classification performances, as both models have similar accuracy scores. In terms of robustness, we observe an increase of about 4% against the FGSM attack for the ACN ensemble and similar robustness scores for both ensembles against the PGD attack. These results demonstrate that it is possible to combine ACN ensembles and adversarial training to further improve robustness to white-box adversarial attacks compared to regular ACN ensembles. However, the robustness gains compared to a regular ensemble with adversarial training are limited to the FGSM attack.

## 6 Limitations

To the best of our knowledge, we are the first to propose an ensemble-based adversarial defense improving ensemble diversity and robustness using a projection network and contrastive learning in a shared latent space. However, we are aware of some limitations with the proposed method such as the limited robustness against strong adversarial attacks such as PGD with regular distortion bounds (0.1 for Fashion-MNIST and 0.031 for CIFAR10) and AutoAttack (see Appendix E). We are also aware that ACN ensembles show better results on CIFAR10 than Fashion-MNIST (see Appendix B and D).

According to Ilyas et al. (2019), neural networks are likely to learn non-robust features. These features are highly predictive and correlated with class labels, but also vulnerable to adversarial attacks. The robustness improvements observed against weaker attacks such as FGSM and PGD with lower distortion bounds might be an indication that non-robust features are diversified in ACN ensembles. This hypothesis is in line with the results obtained in Yang et al. (2020) for DVERGE, an ensemble-based defense in which non-robust features are diversified on purpose. Indeed, the ensemble robustness of the latter defense against strong white-box attacks is also limited. The diversification of non-robust feature could also be a reason as to why ACN ensembles perform better on a more complex dataset such as CIFAR10, which possibly contains more non-robust features due to its greater complexity compared to Fashion-MNIST. However, further experimentation with different datasets are required to confirm the this hypothesis. Addressing those limitations in further

work by improving the training method or testing with different architectures could allow the proposed method to compete against current state-of-the-art adversarial defenses by promoting the diversification of more robust features.

## 7 Conclusion

We introduced ACN ensembles as an adversarial defense based on a novel feature diversification strategy.[1] Our approach leverages notions of contrastive learning and adapts concepts previously used in domain adaptation to diversify the feature representations learned by the different models in an ensemble directly from their feature maps.

We demonstrate experimentally that the proposed method is functional. Specifically, ACN ensembles are compared to regular ensembles in terms of adversarial robustness against popular adversarial attacks and ensemble diversity with a metric based on the transferability of adversarial examples among individual models. The results demonstrate that ACN ensembles are more robust against the FGSM attack and PGD attacks using low distortion bounds. They also show lower transferability of adversarial examples among individual models in ACN ensembles, suggesting that the proposed method promotes the learning of more diverse feature representations.

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
