# OpenReview forum: "Improving Robustness and Diversity with Adversarial Contrastive Network Ensembles"
_TMLR — Rejected by TMLR_

### Review · Reviewer_94G9 · 2023-10-18

**Summary Of Contributions:**

This paper proposes a new training paradigm to boost the robustness of neural ensembles. The paper argues that the adversarial vulnerability of neural ensembles might be rooted in them learning a similar, non-robust set of features. As such, the paper aims to enhance the diversity of feature representations in neural ensembles using a method dubbed Adversarial Contrastive Network (ACN). Apart from the usual ensemble network, ACN introduces a new latent space called the shared latent space by projecting the feature space representation of neural ensembles. As the ACN's goal is to promote diversity, this projection head tries to diversify the features used by each model for each class so that they each ensemble uses a different set of features to represent each class. To train such ensembles, the paper uses a combination of three loss functions: 1) A regular cross-entropy loss for the ensemble, 2) a contrastive loss that aims to pull the same classes in the shared latent space together while pushing different classes apart, and 3) a diversity loss that aims to push the similar classes apart. Experimental results on Fashion-MNIST and CIFAR-10 using ResNet-20 and ResNet-18 models demonstrate that ACN ensembles are more robust compared to vanilla ensembles.

**Audience:**

No

**Broader Impact Concerns:**

Not applicable, even though I encourage the authors to write a broader impact section.

**Claims And Evidence:**

No

**Requested Changes:**

### Questions:

1. Why all the operations used in contrastive and diversity loss functions cannot be done in the original feature space? Do we have empirical evidence that it wouldn't work?

2. As the contrastive and diversity loss are in contradiction with each other, how stable is the whole training process? What tricks have been done to ensure stability of training?

3. In the experimental results, why a subset of the test set is used for evaluation? Given the small scale of CIFAR-10, I hope that the experiments can still be run on the entire test set.

4. In Figure 2, what happens after introduction of the diversity loss? In other words, how does the t-SNE plot change?

5. Why has the number of epochs required to train a classifier on CIFAR-10 to be that high (1000 epochs as per Section 4)? Usually 120 epochs are enough to train ResNet-18 on CIFAR-10 with SGD.

6. To run adversarial training in Section 5, it is mentioned that: "That is, at every epoch during training, adversarial examples are generated for each individual model from the images in the mini-batch. The images generated for each model are then used to compute their individual cross-entropy loss, which are then used to determine the ensemble loss with Eq. 1." If the target is to make the ensemble more robust, why can't we generate adversarial examples against the ensemble rather than individual models?

7. On a general question regarding the paper's results, I was thinking why the ACN ensembles yield significantly better results against FGSM and not PGD attacks. One possible conclusion that I reached was based on the current literature on optimality of FGSM attacks, such as Andriushchenko and Flammarion, 2020 and Jia et al., 2022 regarding fast adversarial training. There is evidence that the quality of the FGSM attack would be sub-optimal if the neural network is locally non-linear. Looking at the ACN ensembles, it seems that the proposed model is encouraging a more locally non-linear behavior, and as such, results in better robustness against FGSM. I wanted to know what the authors or other reviewers think about this explanation?

Andriushchenko and Flammarion, 2020. "Understanding and improving fast adversarial training." _NeurIPS_.

Jia et al., 2022. "Boosting fast adversarial training with learnable adversarial initialization." _IEEE Transactions on Image Processing_.

### Major Suggestions:

- As mentioned above, the paper needs to be better motivated. At the moment, the motivation of increasing diversity leading to more robust features seems not well-supported.

- The paper needs to be way more thorough in its experimental results. As mentioned above, many baselines needs to be added. Also, just evaluating for two small scale datasets such as Fashion-MNIST or CIFAR-10 seems inconclusive. I suggest adding more large scale experiments on ImageNet. Also, for evaluation, it would be better to use AutoAttack (Croce and Hein, 2020.) seems a stronger baseline.

- Systematic ablation studies to quantify the role and importance of each component in its final performance is also crucial.

Croce and Hein, 2020. "Reliable evaluation of adversarial robustness with an ensemble of diverse parameter-free attacks." _ICML_.

### Minor Suggestions:

- Instead of using double indices such as ${\mathbf{q}\_{m, i}}$ using a superscript notation such as ${\mathbf{q}\_{m}^{(i)}}$ would make it easier to follow the rather complicated formulations given in Eqs. (2) and (3). Also, it would be better to use $\boldsymbol{q}$ (with \boldsymbol) instead of $\mathbf{q}$ (with \mathbf) as the second one is often used for stochastic variables. The same thing can be applied to almost all the variables in the paper as they are all deterministic rather than probabilistic.

- In Figure 1, it would be better to add a few lines of explanation to the figure caption and explain the figure and the different colors used.

- I believe that having a figure explaining the contrastive and diversity losses of Eqs. (2) and (3) is also necessary. You can use the push and pull analogy in a figure with images to demonstrate which samples are pulled together and which are pushed away.

- Line 2 of Section 5: "one of the most efficient" -> "one of the most effective". This is very important as adversarial training is no way the most efficient method.

**Strengths And Weaknesses:**

### Strengths:
- The paper is generally well-written, even though it would benefit from a few adjustments given bellow.
- The problem statement is interesting and could be further investigated.

### Weaknesses:
- The most major issue of this submission is its motivation. The paper mentions on several occasions that: "different neural networks trained on the same classification task tend to learn similar feature representations leading to the models having similar adversarial vulnerabilities (Ilyas et al., 2019; Pang et al., 2019)." While it is true that Ilyas et al., 2019 mention this finding in their paper, but the main argument of that paper is the existence of a robust set of features against non-robust ones. In particular, Ilyas et al., 2019 argue that vanilla training tends to heavily rely on non-robust features and this is their source of vulnerability. It is true that neural networks tend to learn similar non-robust features, but there is no guarantee that diversification of features, as argued in this paper, would be enough to address that. In other words, using the current arguments we cannot necessarily say *diversity of features would result in learning robust models*. This can also be seen from the experimental results, where the attack only performs acceptably against FGSM, which is a weak attack.

- The other issue with the approach is the lack of systematic introduction of ideas and ablation studies. The paper uses a different projection space to introduce the bulk of its ideas and loss functions, but I feel one big question that remains unanswered is whether introduction of this space is necessary after all. In particular, why all the operations used in contrastive and diversity loss functions cannot be done in the original feature space? Do we have empirical evidence that it wouldn't work?

- Regarding the experimental results, I think that the baselines used are not sufficient. The paper does discuss the various prior work with similar ideas around adversarial robustness of ensembles, and I believe those methods should also be incorporated in the experimental results as baselines. Moreover, some experimental settings are trivial, and not much insight can be taken from them. For instance, it is widely known that as we decrease the adversarial perturbation radius, we are decreasing the attack strength. As such, ACN performing better against lower norm attacks is trivial. Based on this performance, we cannot conclude that: "The robustness improvements observed against weaker attacks such as FGSM and PGD with lower distortion bounds might be an indication that non-robust features are diversified in ACN ensembles."


Ilyas et al., 2019. "Adversarial examples are not bugs, they are features." _NeurIPS_.

Pang et al., 2017, "Improving Adversarial Robustness via Promoting Ensemble Diversity." _ICML_.

---

> ### Author Response · Authors · 2023-11-14
>
> We thank the reviewer for their detailed review, we appreciate the efforts and level of feedback provided. Here are answers to the various questions and suggestions made.
>
> *Why all the operations used in contrastive and diversity loss functions cannot be done in the original feature space?*: The contrastive loss is tightly associated with the projection network; it is the main loss used to train that network. It would not make much sense to have a contrastive loss in the original feature spaces (note the plural here, as there is one feature space per individual model), the purpose of such loss is not clear without the projection network. Applying the diversity loss directly on the feature spaces is possible, although not effective, as these feature spaces are likely not well aligned together, although the same features can be extracted in a variety of ways. The idea of using a projection network is in fact to achieve that, to align the various feature spaces in a common space, from which we can enforce some diversity between these models. We refer out to the previous answer given to reviewer Un79 for some more explanations showing how the approach is performing.
>
> *How stable is the whole training process?*: To achieve a good stability, we rely on a $\lambda$ parameter to weigh the importance of diversity loss when updating the parameters. Results showing the value of the different loss function for the different models trained in the main papers are presented in section C of the supplementary material. We use a fixed $\lambda$ value for our experiments, but some tricks could have also been used, such as a schedule on the $\lambda$ parameters.
>
> *In the experimental results, why a subset of the test set is used for evaluation?*: A subset of the test set was used because some adversarial attacks such as PGD and AutoAttack can take a long time to execute. The experiments can be done on the entire test set.
>
> *In Figure 2 (now Fig. 3), what happens after introduction of the diversity loss?*: We are not sure whether we understand this question well. We can say that Fig. 2a (now Fig. 3a) present the t-SNE visualization of data in the three feature spaces of the individual models – observe that in each class, there are three clusters – while Fig. 2b (now Fig. 3b) presents the mapping in a shared space with the projection network, where instances of the same class but coming from different models are relatively well grouped.
>
> *Why has the number of epochs required to train a classifier on CIFAR-10 to be that high?*: We used such a high number of epochs as the robustness against  FGSM attack for the ensembles trained on CIFAR10 was still increasing (see Fig. 3 (now Fig. 4) of the main article).
>
> *If the target is to make the ensemble more robust, why can't we generate adversarial examples against the ensemble rather than individual models?*: The adversarial examples can indeed be generated on the whole ensemble instead of the individual models. We decided to generate them on the individual models because we think it makes more sense in this setting, and such a setting is harder than generating a single adversarial sample for the whole ensemble. Indeed, if we generate adversarial instances on the whole ensemble, only 2 out of the 3 models need to be fooled in order to fool the ensemble. Our method ensures that each model is trained with specific adversarial examples fooling the individual models in particular, instead of adversarial examples that may not even fool all models making the ensemble. In our opinion, as long as we compare to a baseline that uses the same method it is okay, and the current setting used in the paper is harder / more conservative.
>
> *It seems that the proposed model is encouraging a more locally non-linear behavior, and as such, results in better robustness against FGSM. I wanted to know what the authors or other reviewers think about this explanation?*: We agree with this hypothesis. The training is probably promoting some locally non-linear behaviors.
>
> *The paper needs to be way more thorough in its experimental results.*: We are aware of the limitation of the proposed method and the fact that adversarial robustness is only improved against FGSM and weaker PGD attacks. Results against AutoAttack are presented in section E of the supplementary material, where we can see that ACN ensembles are as vulnerable to this attack as the vanilla ensemble (see Table 7). This is why we do not claim in the article that our method improves adversarial robustness against such attack and why we mention it in the limitations section of the main paper.
>
> *Systematic ablation studies to quantify the role and importance of each component in its final performance is also crucial.*: We refer to our answer provided to reviewer Un79 on a similar question on ablation of the contrastive and diversity losses.
>
> We agree with all minor suggestions and the corresponding updates were made in the paper.

---

### Review · Reviewer_Un79 · 2023-10-20

**Summary Of Contributions:**

The paper demonstrates an interesting application of contrastive loss to improving adversarial robustness of ensemble models. In the approach, learned representations by different instances of an ensemble model are projected to a common latent space by a projection network. A diversity loss encourages the diversity of representations learned to improve adversarial robustness while a contrastive loss that trains the projection network to align feature maps of the same label. The work studies the adversarial robustness of the proposed approach on CIFAR-10 and Fashion-MNIST datasets using PGSD and FGS adversarial attack

**Audience:**

Yes

**Broader Impact Concerns:**

As the a methodology work on improving models robustness against adversarial attacks, the paper doesn't have a broader impact statement and I have no concerns over the broader and ethical implications of the work.

**Claims And Evidence:**

Yes

**Requested Changes:**

- The number of models ***M*** is an important hyper-parameter for ensemble models. For the completeness of experimental study, the authors should make comparisons and report results based on different choices of ***M***.
- Despite the clear motivation of using the contrastive loss, it's not clear what actual impacts the proposed contrastive loss has on the model's adversarial robustness and whether it is essential for the following reasons: 1) The adversarial robustness of ensemble models primarily comes from the diversity of learned representations and this is encouraged by the diversity loss not the contrastive loss; 2) different from many existing works of contrastive learning, the proposed contrastive loss encourages features with the same label to be similar instead of features from different augmentations of the same input and it is arguable that the classification loss could have similar effects and is also competing against the diversity loss. The work should consider including ablation study results without the contrastive loss or the diversity loss to provide better insights into the importance of the two training objective components.
- There are existing works that studied contrastive learning in the context of adversarial machine learning and the authors can consider including them in related works:
    - Kim, Minseon, Jihoon Tack, and Sung Ju Hwang. "Adversarial self-supervised contrastive learning." Advances in Neural Information Processing Systems 33 (2020): 2983-2994.
    - Fan, Lijie, et al. "When does contrastive learning preserve adversarial robustness from pretraining to finetuning?." Advances in neural information processing systems 34 (2021): 21480-21492.

**Strengths And Weaknesses:**

### Strengths
- The application of contrastive loss in ensemble models to align similar feature maps and improve model's adversarial robustness is novel with a clear motivation. This motivation is further supported by qualitative study results.
- Experiment results show that ensemble models trained with the proposed approach demonstrate better adversarial robustness over vanilla ensemble models in most settings without significantly compromising classification accuracy on clean images. Experiment settings are comprehensive and covers different aspects of the approach including the diversity of learned representations based on transferability of adversarial images between models and the robustness of models under adversarial training.
- The work is very readable and clearly presents the approach with technical details, like the normalization of projected feature maps. The inclusion of pseudo code and algorithm box is appreciated.

### Weakness
- Experiment results in select settings do not show improved accuracy over vanilla models especially when the adversarial attack method is PGD.
- Please see ***Requested Changes*** for other weakness points and associated requested changes.

---

> ### Author Response · Authors · 2023-11-14
>
> We are grateful to the reviewer for the good report, it provides a good summary of the topic and contributions of our paper and appears to appreciate it. As for the requested changes, here are our answers.
>
> *Vary the number of models used in the ensemble*: That’s a good point. In the preliminary experiments in preparing the paper, we varied the size of the ensembles – nothing very formal that can be presented in a paper – and it appears that it has little effect on results (accuracy and robustness) and conclusions that can be drawn, while requiring long trainings and more GPU memory because of the increased complexity of the ensembles and the fact that the models need to be trained in parallel given the loss function used. For the sake of brevity and the fact that it would require time to produce all these results while, apparently, providing little to no insights on the proposal, we think that the results with M=3 are sufficient to support our proposal.
>
> *Effect of contrastive and diversity losses*: The contrastive and diversity losses are working hand in hand to achieve the wanted results. The diversity loss is a key element to ensure that  models making the ensemble have different feature spaces, which is a key element of our proposal to ensure robustness. However, it is possible to obtain feature spaces that seem different, but in fact are equivalent up to some simple transformations. That’s where the projection network and contrastive loss are required, the projection network being used to remap / align the various feature maps on the same space, trained with the contrastive loss. Such an approach ensures a competition at training time between the projection network trained with contrastive loss, trying to map the feature maps in the shared latent space, and the ensemble models trained with diversity loss in order to obtain different features from each individual model. We don’t think it is relevant to study these two elements independently given this design.
>
> Furthermore, results presented in section C of the supplementary material show the values of the two losses during the training of the different ACN ensembles (Fig. 8, now Fig. 9). From these results, we can see the importance of a good balance between the two losses, which is controlled with the lambda parameter. For instance, in Fig. 3b (now Fig. 4b) of the main paper, we see that there is a drop in robustness against FGSM between C-ACN_0.075 and C-ACN_0.1. This drop can be explained by the fact that when too much weight is put on the diversity loss ($\lambda=0.1$ for CIFAR10), the contrastive learning task of the projection network becomes more challenging to learn. This is resulting in a rapid increase of the contrastive loss and a decrease of the diversity loss, as shown in Fig, 8b and 8c (now Fig. 9b and 9c).
>
> *Related works on contrastive learning*: We agree, we added a paragraph in the related work section of the paper with references to the papers of Kim et al. (2020) and Fan et al. (2021).

---

### Review · Reviewer_i2cb · 2023-10-30

**Summary Of Contributions:**

The paper proposes an ensemble-based approach to defend against white-box adversarial attacks. The key contribution is a strategy to explicitly increase the diversity of the models in the ensembles. This is achieved by projecting the output feature maps into a common latent space where contrastive and diversity losses are applied to encourage the models to be adversarially robust.

Experiments are performed on Fashion-MNIST and CIFAR-10, where the proposed method is compared to baselines against FGSM and PGD attacks.

**Audience:**

Yes

**Broader Impact Concerns:**

There are no ethical implications for this work, so there is no need for a Broader Impact Statement.

**Claims And Evidence:**

No

**Requested Changes:**

- Current adversarial defenses are usually evaluated on strong attacks such as AutoAttack. The current paper only evaluates adversarial robustness against weak attacks like FGSM and PGD. Furthermore, Table 2 and Table 6 show that, compared to the baseline, the proposed method is ineffective against a stronger attack like PGD. Evaluating the proposed method against stronger attacks like AutoAttack would help assess the paper's primary claim.
- Several factors have been shown to affect white-box adversarial attacks significantly. This includes data augmentation or use of synthetic data [1] and network architecture [2,3]. Suppose the proposed diversification idea demonstrates improvements across training datasets (i.e., with and without synthetic data) and architectures (standard WRN and robust networks). In that case, it will go a long way in validating the claims made in the paper.
- While t-SNE is commonly employed for visualizing high-dimensional features, t-SNE has an inherent limitation. If the intrinsic dimensionality of the features is larger than 3 (which might be the case here), one could draw incorrect conclusions from t-SNE since we are only looking at a 2D projection of a higher-dimensional object. To this reviewer, the conclusions from Fig. 2 are not very reliable. Perhaps the effectiveness of the contrastive loss can be established through an appropriate quantitative metric.
- A controlled study can be done where the diversity of the networks in the ensemble can be explicitly controlled. Several such networks can be trained for each level of diversity, and each network's adversarial robustness can be evaluated. Then, you can plot adversarial robustness vs diversity. The correlation in the points can be measured through metrics like Spearman or Kendall Tau correlation. This would help establish a relationship between diversity and adversarial robustness. Once the relation is established, methods to explicitly encourage diversity would be helpful. From the results, this reviewer suspects that the correlation is weak, i.e., larger diversity in the feature space does not necessarily improve adversarial robustness.

[1] Better Diffusion Models Further Improve Adversarial Training, ICML 2023

[2] Revisiting Residual Networks for Adversarial Robustness, CVPR 2023

[3] Robust Principles: Architectural Design Principles for Adversarially Robust CNNs, BMVC 2023

**Strengths And Weaknesses:**

- Strengths
  - Explicitly promoting diversity within an ensemble of models is exciting and could be of general interest even beyond adversarial robustness.

- Weaknesses
  - The experimental results do not fully validate the paper's primary claim, namely that enforcing explicit diversity among the networks in the ensemble improves adversarial robustness. The results of robustness against adversarial attacks are mixed. While the proposed method does afford improvements in some cases, especially for simpler attacks like FGSM (Tables 2 and 6) and for larger values of $l_{\infty}$ Norms (Table 3) than the standard values.
  - Most current defenses against adversarial attacks are typically evaluated against stronger attacks, such as AutoAttack. However, this is missing from the paper. Validating the claim would necessitate such an evaluation. Otherwise, the claim needs to be made more specific.
  - Overall, the paper does not establish a clear relation between the diversity of networks in the ensemble and its effect on adversarial robustness. For instance, does low diversity always lead to lower adversarial robustness?

---

> ### Author Response · Authors · 2023-11-14
>
> We thank the reviewer for his evaluation. We are extracting two main requests from the review, that is: 1) the proposal should be evaluated against stronger attacks, namely AutoAttack, and 2) the relation between the level of diversity and the corresponding robustness should be accessed, to better support the claims.
>
> *Evaluation with AutoAttack:* We already presented such results in the supplementary material provided with the paper, see Section E and Tables 7 and 8 of that document. It shows slight improvements of our proposed ACN over vanilla (baseline) ensembles, although such gains are rather small compared to the gains observed with FGSM and PGD. The conclusions of our paper have been devised by having such results in mind.
>
> *Relation between diversity and robustness:* The relation between diversity and robustness has already been presented in Figure 3 (now Fig. 4) of the paper, where each line represents an ACN ensemble with a different $\lambda$ value (+ vanilla ensemble). As a recall, lambda is the diversification parameter and it controls the weight of the diversity loss during the update of the parameters of the feature extraction layers. The higher the value of lambda is, the more weight is put on the diversity loss. We can see on Figure 3 (now Fig. 4) that robustness increases as the value of $\lambda$ increases with a peak at $\lambda=0.02$ and $\lambda=0.075$ for Fig. 3a (Fashion-MNIST) and Fig. 3b (CIFAR10) (now Fig. 4a and 4b), respectively, before dropping for higher values. Such results indicate an apparent relationship between diversity and adversarial robustness. We observe a similar behavior when assessing average transfer robustness with FSGM – not presented in the paper and supplementary material, but it can be added if deemed relevant. We argue that such results are more relevant to support our claims with our proposed approach, acting directly on the diversity loss being shown to have a direct effect on the robustness. This is indeed more appropriate than looking at output diversity correlation proxy measures (e.g., Spearman or Kendall Tau), as these are capturing a different kind of diversity than the one we are relying on in ACN, which is based on the capacity to map together the feature space induced by the networks making the ensemble.
>
> Other points raised by the reviewer:
>
> *Data augmentation (other factors affecting attacks):*  In section 5 of the paper, we present the results for an experiment where ACN ensembles are combined with adversarial training, which is state-of-the-art for improving robustness to adversarial attack that can be viewed as a data augmentation method. Results of the paper show a small improvement against the FGSM attack and no improvement against the PGD attack in comparison to a regular ensemble with adversarial training. We agree that testing our method over different architectures and data augmentation strategies can allow some further validation of our method. But we don’t see other obvious and relevant approaches that may provide us with further essential insights on the method over those provided in Sec. 5 with adversarial training.
>
> *t-SNE for visualizing high-dimensional features*: Thanks for pointing this out. We agree that concluding only based on t-SNE plot is perilous at best and should be backed by further quantitative results. However, we think the current Fig. 2 (now Fig. 3) is useful to illustrate the idea and provide intuition to the reader on how the model proposed is behaving. As for validating the effectiveness of contrastive loss, we think that results presented in Sec. 4.1.2 support the claims on diversity.

---

### Decision · Action_Editor_R4JY · 2023-11-30

**Recommendation:** Reject

**Comment:**

This paper proposed Adversarial Contrastive Network (ACN) ensembles for improving adversarial robustness. While obvious gains are observed against simple adversarial attack methods, such as FGSM, many reviewers pointed out a common concern of lacking convincing and supporting robustness gains against advanced attack methods, such as PGD and AutoAttack, as also acknowledged by the authors in the Limitations section. Another major concern that reviewers raised is the lack of comparisons to related work (defenses) using model ensembles.

The authors' rebuttal only partially addressed some reviewers' concerns, and most reviewers suggested not accepting the paper in its current form due to the two concerns above. I hope the review comments and suggestions are constructive for the authors to prepare the future version.

**Audience:**

In its current form, the paper needs significant revision and updates.

**Claims And Evidence:**

1. Lack of supporting evidence against advanced attack methods
2. Lack of comparisons to related work (defenses) using model ensembles

**Resubmission Of Major Revision:**

The authors may consider submitting a major revision at a later time.